# A Cotton Fabric Composite with Light Mineral Oil and Magnetite Nanoparticles: Effects of a Magnetic Field and Uniform Compressions on Electrical Conductivity

**DOI:** 10.3390/mi14061113

**Published:** 2023-05-25

**Authors:** Gabriela-Eugenia Iacobescu, Madalin Bunoiu, Ioan Bica, Paula Sfirloaga, Larisa-Marina-Elisabeth Chirigiu

**Affiliations:** 1Department of Physics, University of Craiova, 200585 Craiova, Romania; 2Faculty of Physics, West University of Timisoara, 300223 Timisoara, Romania; 3Advanced Environmental Research Institute, West University of Timisoara, 300223 Timisoara, Romania; 4Condensed Matter Department, National Institute for Research and Development in Electrochemistry and Condensed Matter, 300254 Timisoara, Romania; 5Department of Pharmacy, “Constantin Brancusi” University of Targu-Jiu, 210185 Targu Jiu, Romania; larisa.chirigiu@asociatiadidactica.ro

**Keywords:** magnetic liquid, magneto-tactile sensor, magnetite nanoparticles, cotton microfibers, magneto deformations

## Abstract

Over the past few decades, tactile sensors have become an emerging field of research with direct applications in the area of biomedical engineering. New types of tactile sensors, called magneto-tactile sensors, have recently been developed. The aim of our work was to create a low-cost composite whose electrical conductivity depends on mechanical compressions that can be finely tuned using a magnetic field for magneto-tactile sensor fabrication. For this purpose, 100% cotton fabric was impregnated with a magnetic liquid (EFH-1 type) based on light mineral oil and magnetite particles. The new composite was used to manufacture an electrical device. With the experimental installation described in this study, we measured the electrical resistance of an electrical device placed in a magnetic field in the absence or presence of uniform compressions. The effect of uniform compressions and the magnetic field was the induction of mechanical–magneto–elastic deformations and, as a result, variations in electrical conductivity. In a magnetic field with a flux density of 390 mT, in the absence of mechanical compression forces, a magnetic pressure of 5.36 kPa was generated, and the electrical conductivity increased by 400% compared to that of the composite in the absence of a magnetic field. Upon increasing the compression force to 9 N, in the absence of a magnetic field, the electrical conductivity increased by about 300% compared to that of the device in the absence of compression forces and a magnetic field. In the presence of a magnetic flux density of 390 mT, and when the compression force increased from 3 N to 9 N, the electrical conductivity increased by 2800%. These results suggest the new composite is a promising material for magneto-tactile sensors.

## 1. Introduction

Composite materials are made up of immiscible components: components that retain their identity within the composite. Each of the components of the newly formed entity has specific physical properties; however, through assembly, the resulting entity acquires physical properties superior to those of the components. In recent decades, many new composites have been developed, some with very valuable properties. Numerous composites have been designed to be good conductors or insulators of heat or to have certain magnetic properties. Such composites are used in a wide range of electrical devices, including sensors.

A typical disadvantage of composites is cost. While manufacturing processes are often more efficient when composites are used, the raw materials are expensive. Thus, manufacturing low-cost sensors for various applications, for example, in the medical field, is a challenging research direction that is continuously evolving [1,2,3,4,5,6].

On the other hand, in recent years, magnetic sensors have been developing in the direction of higher sensitivity, lower power consumption, smaller size, etc. [7]. The performances of some types of magnetic sensors have already met the requirements of tactile sensing technology [8]. Therefore, a new type of tactile sensor, called a magneto-tactile sensor, has recently been developed [9,10,11]. Compared with tactile sensors based on other mechanisms, magneto-tactile sensors have the advantages of high sensitivity, low hysteresis, low power consumption, and easy implementation in three-dimensional and remote sensing applications [12]. In particular, magnetic composites (MCs) are manufactured using natural and/or artificial fibers mixed with magnetizable liquids and/or ferro-ferrimagnetic nano-microparticles. Thus, in [13], MCs were obtained from cotton microfibers combined with magnetorheological suspensions based on silicone oil with carbonyl iron (CI) microparticles. In MCs made in this way, the components of the complex dielectric permittivity can be modified in a magnetic field, a property that makes them useful in sensors. X-band microwave absorbers have been made [14] using MCs based on CI microparticles and graphite nanoparticles. Microwave absorbers with frequencies between 8 and 12 GHz were created [15] using microparticles of CI with a surface coated with a layer of copper and fabrics based on cotton microfibers. Research on similar topics was reported in [16], where cotton fabrics were coated with αFe_2_O_3_ nanoparticles, graphene nanoparticles, and iron nanoparticles. In [17,18], polymer fibers were treated with BaFe nanoparticles and toner type 6754CP-313, and in [19], with liquid solutions of polyurethane polyole and microparticles of the NdFeB type. Therefore, composites made in [13,14,15,16,17,18,19] have physical properties that make them useful in electromagnetic smog protection.

Recently, research has been carried out with the aim of making environmentally friendly MCs. In [20], MCs were manufactured from cotton microfibers, honey, CI microparticles, and turmeric powder. In these composites, the complex relative dielectric permittivity components were fixed based on the quantitative ratio between bee honey and turmeric powder for the same amount of CI microparticles. For the same competition, the relative and complex dielectric permittivity components were sensitively modified in a magnetic field, a property that can be used in the creation of medical devices.

In the present study, we proposed a newly manufactured MC based on cotton fabric (GB) and magnetic liquid (LM). The GB fabric had dimensions of 30 × 30 × 0.4 mm^3^ and was made of 100% cotton fibers. The LM liquid was based on light mineral oil and magnetite nanoparticles (Fe_2_O_3_). An electrical device, ED, was created to study the electrical conductivity of the MC under mechanical and magnetic stress. Using the experimental setup described in the experimental section, the dependence of the electrical conductivity of the MC was determined as a function of the *p* values of the uniform mechanical compression force and, respectively, as a function of the *B* values of the magnetic flux density. We observed that the values of the electrical conductivity of the MC increased with an increase in the *p* values of the mechanical compression. At constant *p* values, the electrical conductivity of the MC was sensitively influenced by the *B* values of the magnetic flux density, which makes them suitable for magneto-tactile sensors fabrication.

## 2. Materials and Methods

### 2.1. Magnetic Composite (MC) Manufacturing

The materials used for the fabrication of the composite were:Cotton fabric (GB), with dimensions of 30 × 30 × 0.4 mm^3^ and a volume of $V_{\mathrm{GB}}=0.36{\;\mathrm{cm}}^{3}$, approximately equal to that of cotton microfibers. The mass of the GB fabric, measured with the help of an analytic balance (AXIS 60 type), was $m_{\mathrm{GB}}=0.372\;g$. The mass density of the GB fabric was $\rho_{\mathrm{GB}}=m_{\mathrm{GB}}/V_{\mathrm{GB}}\approx \;1.033g/{\mathrm{cm}}^{3}$.Magnetic liquid (ML), type EFH-1, was produced by Ferrotec (Santa Clara, CA, USA) [21] and bought from Magneo Smart (Sendreni, Romania) [22]. The ML was based on light mineral oil from Sigma-Aldrich Chemie GmbH (Taufkirchen, Germany) number CAS 8012-95-1, with standard density $0.873\;g/{\mathrm{cm}}^{3}$ and viscosity 25−80 mPa×s at 20 °C, and magnetite nanoparticles (${\mathrm{Fe}}_{3}O_{4}$). ML had the following technical characteristics:○A mass density of $\rho_{\mathrm{ML}}\;≃1.21\;g/{\mathrm{cm}}^{3}$;○An average diameter of the particles of $d_{{\mathrm{Fe}}_{3}O_{4}}=11.6\;\mathrm{nm}$ [23];○A saturation magnetization of about $M_{S_{\mathrm{ML}}}\;≃35\;\mathrm{kA}/m$;○An ignition temperature $t_{\mathrm{ign}}\approx 91\;{}^{\circ}C$;○A volume fraction of ${\mathrm{Fe}}_{3}O_{4}$ of $\Phi_{{\mathrm{Fe}}_{3}O_{4}}=6.5\;\mathrm{vol}.\%$, as reported in [24].

The manufacturing of MCs was performed via the following four steps:

Step 1: A quantity of ML with the volume $V_{\mathrm{ML}}=2{\;\mathrm{cm}}^{3}$ was introduced into a Berzelius glass with the GB fabric.

Step 2: The Berzelius glass, containing the ML liquid and the GB fabric, was heated until the temperature of the ML liquid, measured with an infrared thermometer type AX-6520 (from AXIO MET), reached *t* ≈ 80 °C.

Step 3: After 24 h, the fabric with the incorporated ML liquid was pressed between two absorbent papers to extract the ML excess. At the end of this stage, the MC was obtained.

Step 4: The mass of MC weighed with the AXIS 60 balance was $m_{\mathrm{MC}}=0.760\;g.$ Then, we could calculate the mass of ML in MC as $m_{\mathrm{ML}}=m_{\mathrm{MC}}-m_{\mathrm{GB}}=0.760\;g-0.372\;g=0.388\;g$.

The volume of ML in MCs was calculated as follows: $V_{\mathrm{ML}}=m_{\mathrm{ML}}/\rho_{\mathrm{ML}}=0.388\;g\;/\;1.21\;g/{\mathrm{cm}}^{3}\approx 0.321{\;\mathrm{cm}}^{3}$.

In a volume of $0.321{\;\mathrm{cm}}^{3}$ of ML, there was 6.5 vol.% of ${\mathrm{Fe}}_{3}O_{4}$ nanoparticles. Consequently, the volume occupied by nanoparticles was $V_{{\mathrm{Fe}}_{3}O_{4}}\approx 0.021{\;\mathrm{cm}}^{3}$.

Using the above-calculated volumes, we estimated the volume of ML as $V_{\mathrm{oil}}\approx V_{\mathrm{ML}}-V_{{\mathrm{Fe}}_{3}O_{4}}=0.321{\;\mathrm{cm}}^{3}-0.021{\;\mathrm{cm}}^{3}=0.300{\;\mathrm{cm}}^{3}$, and the volume of cotton microfibers as $V_{f}=V_{\mathrm{GB}}-V_{\mathrm{ML}}=0.360{\;\mathrm{cm}}^{3}-0.321{\;\mathrm{cm}}^{3}=0.039{\;\mathrm{cm}}^{3}$.

Therefore, the MC volume is $V_{\mathrm{MC}}=V_{f}+V_{\mathrm{oil}}+V_{{\mathrm{Fe}}_{3}O_{4}}=0.360{\;\mathrm{cm}}^{3}$. This result led us to the following value for MC thickness: $h_{\mathrm{MC}}=V_{\mathrm{MC}}/S_{\mathrm{MC}}=0.360{\;\mathrm{cm}}^{3}/9{\;\mathrm{cm}}^{2}\approx 0.040\;\mathrm{cm}$, same value as the GB thickness, as expected after the Step 3 of manufacturing.

With the values $V_{\mathrm{MC}},V_{\;\mathrm{GB}},V_{\mathrm{oil}}\;si\;V_{{\mathrm{Fe}}_{3}O_{4}}$, we can calculate the volume fractions of the microfibers ($\Phi_{f}$), oil ($\Phi_{\mathrm{oil}}$), and magnetite microparticles ($\Phi_{{\mathrm{Fe}}_{3}O_{4}}$), as shown in Table 1.

### 2.2. Electrical Device (ED) Manufacturing

For manufacturing the electrical device ED, the following materials were used:A simple textolite plate (PCu), coated with copper on one side and with dimensions of 100 × 75 × 0.8 mm^3^, from Electronic Light Tech (Bucharest, Romania) [25]. The PCu was based on an epoxy resin (FR4 type) reinforced with fiberglass, and one face was covered with a 35 μm thick copper layer.An MC with dimensions of 30 × 30 × 0.4 mm^3^ and the components from Table 1.A hypoallergenic patch on silk support type Omniplast, bought from S. C. Hartmann S. R. L. (Bucharest, Romania) [26]. The patch was a self-adhesive tape with a width of 5 cm, thickness of 0.22 mm, and length of 20 m. It was high-temperature- and wear-resistant. We denoted this patch as SAT.

The main stages for the fabrication of the ED were:From the PCu, we cut two identical pieces, each with dimensions of 30 × 30 × 0.8 mm^3^.On the copper side of each plate, two copper conductors were attached via hot welding.The PCu plates and MC were arranged in a sandwich-type structure so that MC was in contact with the copper sides of two PCu boards.Strips with dimensions 10 cm × 2.5 cm were cut from the SAT, with which the assembly was reinforced. Through this consolidation, electrical contact was made between the copper foil of the PCu and MC. At the end of this stage, the ED device was obtained.

### 2.3. Experimental Methods and Measurements

The visualization of the GB fabric was performed using a digital microscope equipped with an image sensor with a resolution of 640 × 480 and a magnification from 25× to 200×. MC structure was visualized with an optical microscope produced by Optika (Ponteranica, Italy). The magnetization, *M*, as a function of the magnetic field intensity, *H*, was recorded with an experimental setup as described in [27]. The surface morphologies and the elemental compositions of the obtained samples, were investigated using a scanning electron microscope Inspect S (SEM) from FEI Europe B.V., Eindhoven, the Netherlands, equipped with an X-ray energy-dispersive spectrometer (EDX). The samples were analyzed in low vacuum mode using an LFD detector with a spot value of 3.5, a pressure of 30 Pa, and a high voltage of 30 kV.

The experimental setup used for the study of the electrical conductivity of the MC under mechanical and magnetic stress is represented in Figure 1.

The experimental setup consisted of a direct current electromagnet, the DCS current source, the Br bridge, the Gs Gaussmeter with the h probe, and a force application unit for the deformation of the ED device. The DCS current source was an RXN-3030D type, produced by HAOXIN (Hefei City, China), the bridge Br was an RLC-meter, type CHY 41R601 (Lodz, Poland), and the Gaussmeter was a DX-102 type, from Dexing Magnet Tech Co. (Xiamen, China).

The ED deformation unit contained non-magnetic elements. The unit consisted of a shaft that passes through the magnetic pole of the electromagnet and is mechanically coupled with a disc and a platter. The ED and the probe h of the Gaussmeter were fixed between the poles of the electromagnet by means of the non-magnetic disc (pos. 5 of Figure 1).

The marked masses (pos. 6 from Figure 1) were made of lead powder. The relationship between the lead masses, the compression (gravity) force, *F*, and the compression pressure, $p_{F}$, exerted on the ED, is shown in Figure 2.

The ED device and the probe h of the Gaussmeter were inserted between the N and S poles of the electromagnet, as shown in Figure 1. During the measurements, the magnetic flux density, *B*, was fixed at values within ±2% error limits. The electrical resistance, *R*, of the ED was measured with the bridge Br about 30 s after fixing the magnetic flux density, *B*, at a certain value.

## 3. Results and Discussion

The GB fabric (Figure 3a) consists of interwoven threads in the form of meshes. Viewed under the digital microscope (Figure 3b) it can be seen that the fabric threads are made of microfibers. There are free spaces between the microfibers; therefore, the ML liquid was absorbed via capillarity.

Inside the MC composite (Figure 4a) the ML liquid is absorbed at the level of the cotton threads. It can be seen from Figure 4b the existence of liquid ML between the microfibers of each cotton thread.

The authors of [29] report that for the EFH1 magnetic liquid, the standard deviation from the average hydraulic diameter of the magnetite nanoparticles is 0.474 ± 0.013. We can consider that in the case of the EFH1-type liquid used in the present paper, the polydispersity of the magnetite nanoparticles does not affect the experimental results. On the other hand, the use of the MC composite manufacturing technology, described above, uniformly distributes the ML liquid in the GB fabric, thus contributing to reproducible results during measurements.

Figure 5a shows the magnetization, *M*, as a function of the magnetic field intensity, *H* [27]. Between the saturation magnetization $M_{S_{\mathrm{ML}}}$ of the magnetic liquid, and the saturation magnetization $M_{S_{\mathrm{MC}}}$ of the composite, the following relationship holds [30]: $\mu_{0}M_{S_{\mathrm{MC}}}=\Phi_{{\mathrm{Fe}}_{3}O_{4}}\mu_{0}M_{S_{\mathrm{ML}}}$, where $\mu_{0}$ is the vacuum magnetic constant, and the values of $\Phi_{n}$ are listed in Table 1. Based on this equation and the data in Table 1, we obtained the MC magnetization plots, as shown in Figure 5b.

From Figure 5b, it can be seen that the magnetization *M* of the MC increased linearly with an increase in *H* up to H=200 kA/m. Above this value of *H*, the increase in *M* was slow. For values H=500 kA/m, the magnetization tended towards saturation.

The SEM image of MCs (Figure 6a) shows the surface morphology of the cotton microfibers with magnetic liquid, and the EDX analysis (Figure 6b) shows the presence of C, O, Fe, and Ca elements, specific to the fabric and the magnetic liquid used.

A quantitative analysis of the elements in the microfiber (in mass and atomic percentages) is presented in Table 2.

The elemental distribution of the Fe, Ca, O, and Al components in the cotton microfibers is shown in Figure 7.

From Figure 7, a quasi-uniform distribution of the elements highlighted by the elemental analysis is observed.

The configuration of electrical device ED, made according to the procedure described in Section 2.2, is shown in Figure 8a,b.

It can be seen from Figure 8b that the ED was a unitary body and resistant to mechanical actions of medium intensity (such as falls and hits). With consolidation under pressure with self-adhesive tape, the device in Figure 8b was obtained, which had no air gaps between the copper foil of the PCu boards and the surfaces of the MC. The electrical contact was considered established when the electrical resistance of the ED terminals, measured with an UT-60 ohmmeter, was finite.

The electrical resistance measurements were performed as functions of the magnetic flux density, *B*, with the weight forces as a parameter, *F*, of the marked masses placed on the disc of the experimental setup from Figure 1, as described in Section 2.3. The obtained experimental results are represented as the dependencies $R=R{\left(F,\;p_{F}\right)}_{B=0\;\mathrm{mT}}$ and $R=R{\left(B\right)}_{F=0N}$ in Figure 9a and Figure 9b, respectively.

It can be seen from Figure 9a that the values of the electrical resistance, *R*, decreased linearly with an increase in force, *F*, along with an increase in pressure, $p_{F}$. From Figure 9b, it can be seen that the values of *R* decreased significantly with an increase in the size of *B*. Thus, for *B* = 0 mT, the value of the electrical resistance was *R* = 1377 MΩ, and it decreased to *R* = 331 MΩ at *B* = 390 mT and $p_{F}=0\;\mathrm{kPa}$, which means 24% of the electrical resistance value remained in the absence of the magnetic field. From Figure 9b, it can be seen that, upon increasing the parameter *F*, the values of *R* decreased as follows: At *F* = 3 N, the electrical resistance had the value *R* = 725 MΩ for *B* = 0 mT and decreased to *R* = 129 MΩ (17.8% of the value at zero field) for *B* = 390 mT. Upon increasing the *F* parameter to 9 N, *R* decreased from *R* = 420 MΩ for *B* = 0 mT to *R* = 51 MΩ (12.14% of the value at zero field) for *B* = 390 mT. In summary, the experimental data show that the electric resistance, *R*, decreased linearly with an increase in *F* and $p_{F}$, but decreased non-linearly with an increase in the magnetic flux density, *B*.

To describe the mechanisms participating in the observed effects, we imagined a cross-section through the ED from Figure 8. The results shown in Figure 9 suggest that the ED was a resistor. The value of the electrical resistance, *R*, could be modified by varying the mechanical compression, *p*, and the magnetic flux density, *B*. Based on this, we calculated the electrical conductivity, σ, from the electrical resistance formula as follows:(1)$$\sigma =\frac{h_{0}}{RL^{2}},$$
where $h_{0}$ is the thickness of the MC, *R* is the electrical resistance, and $L^{2}$ is the area of the common electrode–MC surface.

From Equation (1), with $h_{0}=4\cdot 10^{-4}\;m$ and $L^{2}=9\cdot 10^{-4}\;m^{2}$, we have the following:(2)$$\sigma \;\left(\Omega^{-1}\cdot m^{-1}\right)=\frac{4}{9\cdot R\left(M\Omega \right)}$$

Using the experimental data $R=R{\left(F,p_{F}\right)}_{B=0\;\mathrm{mT}}$ from Figure 9a, we introduced in Equation (2) the functions $\sigma =\sigma {\left(F,p_{F}\right)}_{B=0\;\mathrm{mT}}$ that were obtained as in Figure 10a. Similarly, using the data $R=R{\left(B\right)}_{F}$ from Figure 9b, introduced in Equation (2), we obtained the functions $\sigma =\sigma {\left(B\right)}_{F}$, as shown in Figure 10b.

As expected, from Figure 10a,b, it follows that the dependence of the electrical conductivity of the MC, σ, on *B* and *F* was inversely proportional to that of the electrical resistance (Figure 9a,b). On the other hand, from Figure 10a,b, it can be seen that, for the same value of the compression force, *F*, when a magnetic field was applied, σ increased significantly with an increase in the magnetic flux density, *B*.

Relevant data extracted from Figure 10 are summarized in Table 3.

Consequently, between the electrical conductivity σ and the compressive stress induced by the force, *F*, and the magnetic field density values, *B*, there was a close connection. In order to describe the mechanisms that contributed to the observed phenomena, we considered the idealized model from Figure 11. According to this model, magnetite nanoparticles have spontaneous magnetization [31] and become electrically charged by friction [32], which will cause agglomeration and micro-ball (NS) formation in the GB fabric. We assumed that the NSs had the same diameter and were uniformly distributed in the fabric. In an external magnetic field, the NSs were instantly magnetized, transforming into magnetic dipoles, $\vec{m}$ (Figure 11b).

The magnetic moment $\vec{m}$ of the NS spheres, projected in the direction of the Oz axis, could be approximated using the following equation [20,33]:(3)$$m=\pi d^{3}\frac{B}{2\mu_{0}},$$
where *d* is the average diameter of the magnetic dipole, *B* is the magnetic flux density, and $\mu_{0}$ is the magnetic constant of the vacuum.

Magnetic interactions took place between two neighboring dipoles, $\vec{m}$, placed in a magnetic field. The magnetic interaction force between these dipoles, projected along the direction of the Oz axis, was calculated with the following equation [20,33]:(4)$$F_{m}=-\frac{2.25\cdot \Phi_{{\mathrm{Fe}}_{3}O_{4}}L^{2}B^{2}}{\mu_{0}},$$
where $\Phi_{{\mathrm{Fe}}_{3}O_{4}}$ is the volume fraction of ${\mathrm{Fe}}_{3}O_{4}$ nanoparticles, $L^{2}$ is the contact surface area of the MC with that of the copper foils, and *B* is the magnetic flux density.

From Equation (4), we obtained the magnetic pressure, denoted by $p_{B}$, in the following form:(5)$$p_{B}=-\frac{2.25\cdot \Phi_{{\mathrm{Fe}}_{3}O_{4}}B^{2}}{\mu_{0}},$$

For $\Phi_{{\mathrm{Fe}}_{3}O_{4}}=7\;\mathrm{vol}.\%$, $\;L^{2}=9{\;\mathrm{cm}}^{2}$, and $\mu_{0}=4\pi \cdot 10^{-7}H/m$ inserted in Equation (4), we obtained the function $F_{m}=F_{m}{\left(B\right)}_{F=0\;N}$ as represented in Figure 12a. In Equation (5), we input $\Phi_{{\mathrm{Fe}}_{3}O_{4}}=7\;\mathrm{vol}.\%$, and $\mu_{0}=4\pi \cdot 10^{-7}H/m$ and obtained the function $p_{B}=p_{B}{\left(B\right)}_{F=0\;N}$, also shown in Figure 12a.

It can be seen from Equations (4) and (5), as well as from Figure 10, that the values of $F_{m}$ and $p_{B}$ induced by the magnetic field in the MCs were significantly influenced by the magnetic flux density, *B*.

As is known [28], in the presence of the magnetic field superimposed on the field of mechanical deformations, at equilibrium, we have the following equation:(6)$$F_{m}+F=F_{\mathrm{rBF}}$$
where $F_{m}$ is the magnetic force, F is the gravity force and $F_{\mathrm{rBF}}$ is the reaction elastic force of MC, with the following:(7)$$F_{\mathrm{rBF}}=-k_{\mathrm{BF}}\left(h_{\mathrm{BF}}-h_{0}\right),$$
where $k_{\mathrm{BF}}$ is the elastic constant of MC placed in magnetic field superimposed on the mechanical deformation field, $h_{\mathrm{BF}}$ and $h_{0}$ are the thicknesses of MC in the presence and, respectively, in the absence of the magnetic field superimposed on the mechanical deformation field.

If in Equation (6) we introduce Equations (4) and (7), we obtain:(8)$$h_{\mathrm{BF}}=h_{0} [1-\frac{1}{h_{0}k_{\mathrm{BF}}}\cdot \left(\frac{2.25\Phi_{{\mathrm{Fe}}_{3}O_{4}}L^{2}B^{2}}{\mu_{0}}+F\right)],$$
where the notations are as outlined above.

Using (8), we can calculate the electric resistance as:(9)$$R_{\mathrm{BF}}=R_{\mathrm{BF}}{}_{0}\cdot [1-\frac{1}{h_{0}k_{\mathrm{BF}}}\cdot \left(\frac{2.25\Phi_{{\mathrm{Fe}}_{3}O_{4}}L^{2}B^{2}}{\mu_{0}}+F\right)],$$
where $R_{\mathrm{BF}}{}_{0}$ is the electric resistance for *F* = 0 N and *B* = 0 mT, and the other notations are as outlined above.

From Equation (9), we calculated the electric conductivity as follows:(10)$$\sigma =\frac{\sigma_{0}}{ [1-\frac{1}{h_{0}k_{\mathrm{BF}}}\cdot \left(\frac{2.25\Phi_{{\mathrm{Fe}}_{3}O_{4}}L^{2}B^{2}}{\mu_{0}}+F\right)]},$$
where $\sigma_{0}$ is the electric conductivity for *F* = 0 N and *B* = 0 mT and the other notations are as outlined above. Here, $k_{\mathrm{BF}}$ depends on the magnetic flux density.

Taking into account the data represented in Figure 10a,b, we obtained, as shown in Figure 12b, the functions $\sigma =\sigma {\left(p\right)}_{F}$, where $p=p_{F}+p_{B}$ is the total pressure exerted on an MC placed in a magnetic field.

From Figure 10b and Figure 12b, we can observe that, in the absence of compression forces and a magnetic field, the electrical conductivity value of the MC was $\sigma \approx 3.11\cdot 10^{-4}\;\Omega^{-1}m^{-1}$. Upon increasing the magnetic flux density to a value of *B* = 390 mT ($p_{B}=5.36\;\mathrm{kPa}$), the electrical conductivity of the MC increased and reached a value $\sigma \approx 13.28\cdot 10^{-4}\;\Omega^{-1}m^{-1}$.

Upon applying the compression force *F*, the electrical conductivity value of the MC increased in the absence of the magnetic field and was amplified with an increase in the magnetic flux density, B. Thus, from Figure 12b, it can be seen that, for *F* = 3 N and *B* = 0 mT, the electrical conductivity of the MC increased and reached a value $\sigma \approx 4.06\cdot 10^{-4}\;\Omega^{-1}m^{-1}$, an increase of ~800%, when the magnetic flux density increased to *B* = 390 mT.

For *F* = 9 N and *B* = 0 mT, the electrical conductivity increased by ~300% compared to the electrical conductivity of MCs at *F* = 3 N and *B* = 0 mT. In addition, with an increase in magnetic flux density at a value *B* = 390 mT and the force at a value *F* = 9 N, the electrical conductivity of MCs increased by ~2800% compared to the composite at *F* = 3 N and *B* = 390 mT.

The obtained results can be explained if we consider the electrical conductivity of MCs to be electronic. The layer of iron oxides and oil formed a potential barrier [34]. The height of the electric potential barrier decreased with an increase in pressure generated by the compressive force field. Upon applying a magnetic field, superimposed on a compression force field, the height of the potential barrier was lowered much further, and the electrons from lower energy levels could pass over the potential barrier. This resulted in an increase in the electrical conduction current, which was equivalent to an increase in the electrical conductivity of the MC.

Similar effects are described in [20,35], where the fabrication of composites based on polymers and CI microparticles is reported. In [20], a composite was made based on natural polymers, and in [35], one based on a synthetic polymer was created. In both cases, the electrical conductivity was electronic. By applying a magnetic field, magneto-elastic deformations were generated in the composites, which reduced the contact distance between the conductive microparticles. This resulted in an increase in the electrical conductivity of the composite with an increase in the *B* values of the magnetic flux density [20] and, respectively, of the contact pressure, as reported in [35]. Compared to the MCs created in the present work, in previous studies, CI microparticles were not sufficiently stable on the polymer microfibers. In the case of the composite in this work (see Figure 2), the magnetite nanoparticles were absorbed into the microfibers, favoring a stable electrical response to external mechanical and magnetic demands.

## 4. Conclusions

This paper reports the manufacturing of a new magnetic composite from cotton fabric impregnated with magnetic liquid based on light mineral oil and magnetite nanoparticles. The magnetic liquid was absorbed into the microfibers, and the component elements (Figure 3b) had a quasi-uniform distribution. An electrical device was manufactured from the magnetic composite and its electrical properties were studied when the device was placed in a magnetic field and subjected to axial mechanical forces. From the electrical resistance measurements, we noticed that the electrical conductivity of the magnetic composite increased linearly under the action of axial forces and was strongly influenced by the presence of a magnetic field. This effect was caused by the increasing proximity of the nanoparticles under the action of axial mechanical forces and was amplified in a magnetic field. The results obtained in the paper are qualitatively described by the dipole magnetic approximation model, and are a starting point for making low-cost electrical devices, such as magneto-tactile sensors, based on fabrics with natural/artificial fibers and magnetic liquids.

## Figures and Tables

**Figure 1 micromachines-14-01113-f001:**
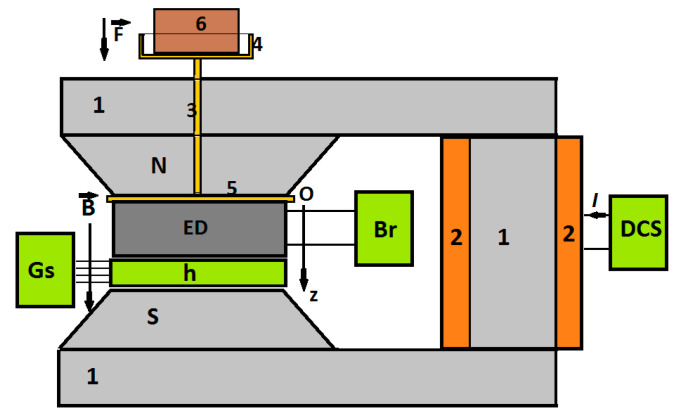
Experimental setup [28]. 1—magnetic core, 2—coil, 3—non-magnetic spindle, 4—non-magnetic plate, 5—non-magnetic disc, 6—marked non-magnetic masses, N and S—magnetic poles, ED—electric device, Br—RLC bridge, Gs—Gaussmeter, h—Hall probe, DCS—continuous source current, Oz—coordinate axis, $\vec{B}$—magnetic flux density vector, $\vec{F}$ —force vector, *I*—intensity of the electrical current.

**Figure 2 micromachines-14-01113-f002:**
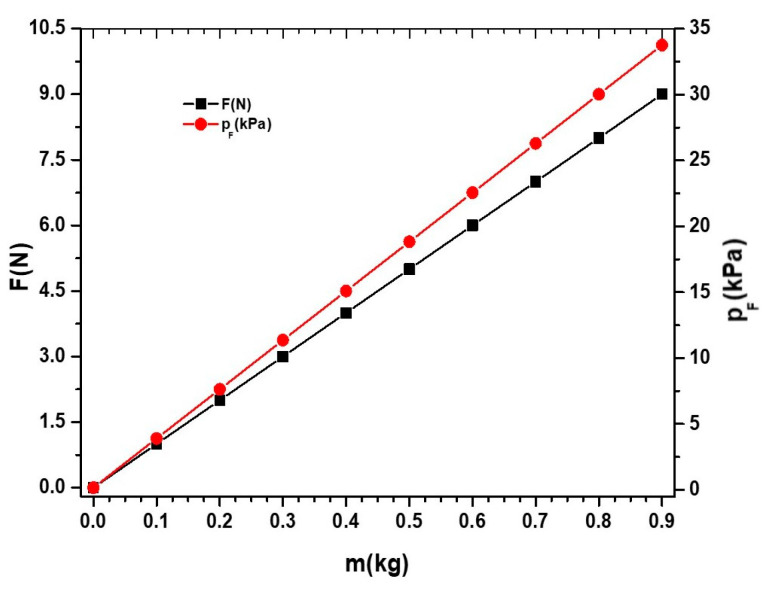
Dependence of force, *F*, and pressure, $p_{F}$, on the mass, *m*, of the disc from the experimental setup.

**Figure 3 micromachines-14-01113-f003:**
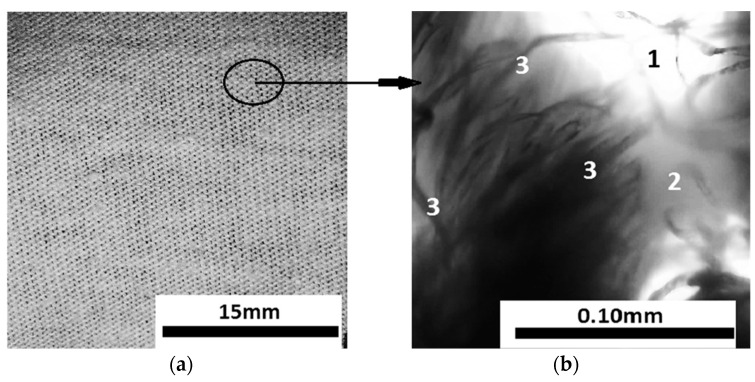
(**a**) GB fabric with dimensions of 30 × 30 × 0.4 mm^3^; (**b**) optical microscopy detail inside the marked area in (**a**); 1—empty space between cotton threads, 2—cotton threads with a diameter $d_{\mathrm{ct}}=0.12\;\mathrm{mm}$, 3—cotton microfibers with a diameter of $d_{\mu f}\approx 5.5\;\mu m$.

**Figure 4 micromachines-14-01113-f004:**
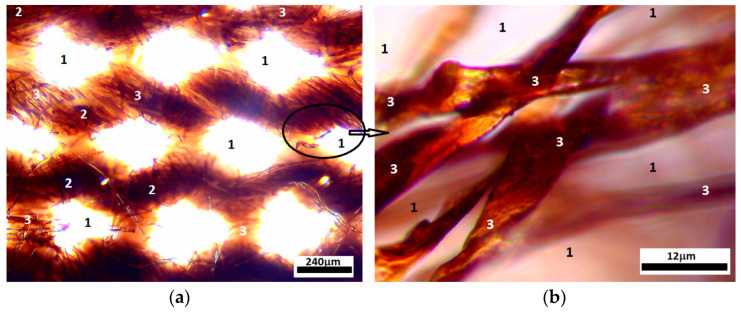
(**a**) MC visualized with the optical microscope; (**b**) optical microscopy detail inside the marked area in (**a**): 1—empty space between cotton threads, 2—cotton threads impregnated with ML, 3—cotton microfibers impregnated with ML.

**Figure 5 micromachines-14-01113-f005:**
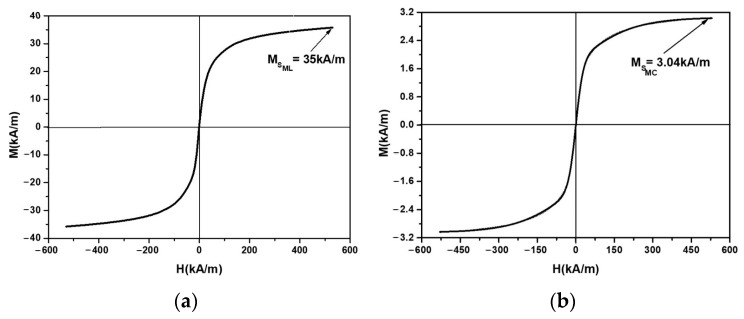
The magnetization, *M*, as a function of the magnetic field intensity, *H*: (**a**) for ML; (**b**) for MC.

**Figure 6 micromachines-14-01113-f006:**
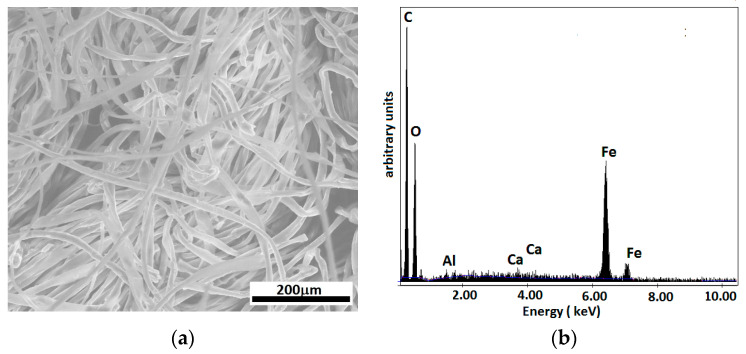
(**a**) SEM image of MC; (**b**) elemental analysis of MCs.

**Figure 7 micromachines-14-01113-f007:**
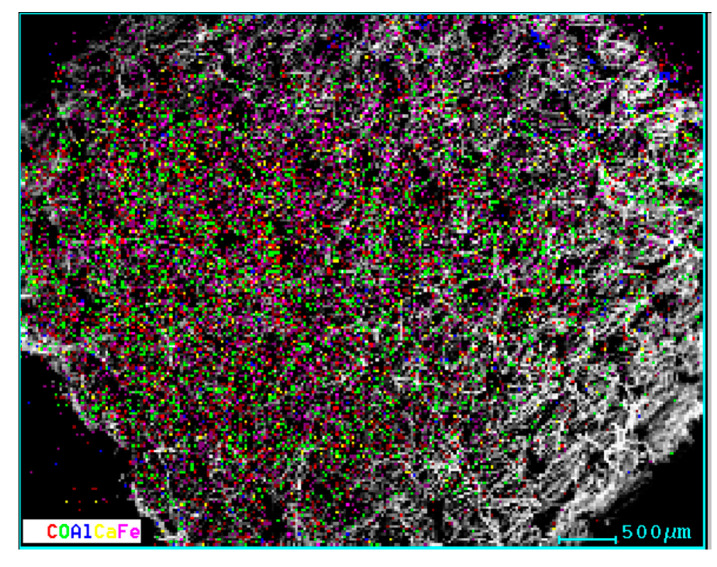
Distribution of elements in the cotton microfiber with magnetic liquid.

**Figure 8 micromachines-14-01113-f008:**
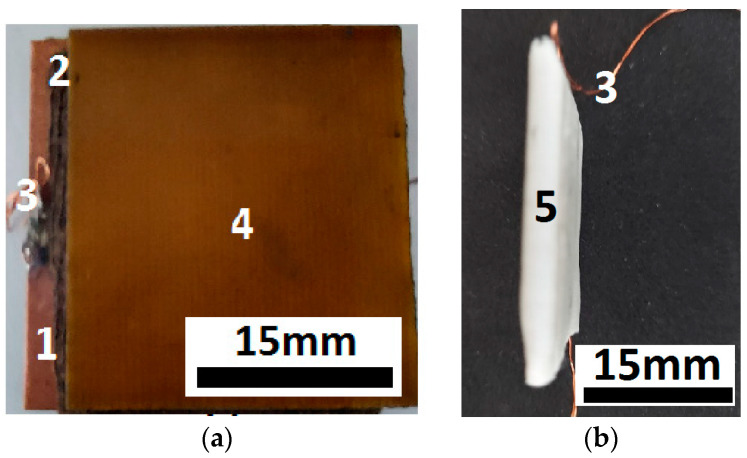
(**a**) PCu–MC–PCu sandwich structure; (**b**) ED consisting of two PCu boards with electrical contacts, among which was the MC consolidated with the SAT (1—copper foil, 2—MC, 3—electrical contact, 4—epoxy resin plate, type FR4 and reinforced with fiberglass, 5—SAT self-adhesive foil).

**Figure 9 micromachines-14-01113-f009:**
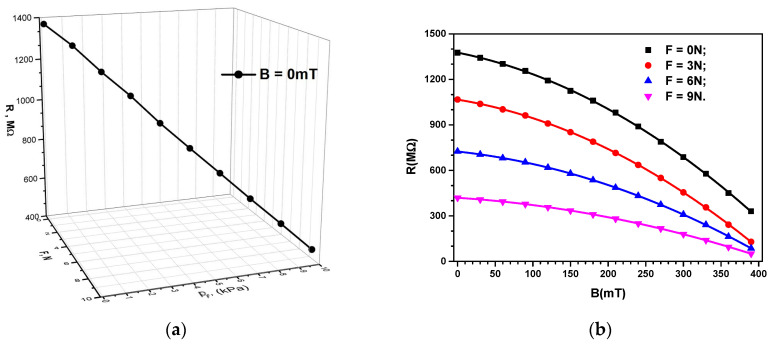
(**a**) The electrical resistance of the ED, *R*, as a function of the compression pressure, $p_{F}$, and the compression force, *F*; (**b**) the electrical resistance of the ED, *R*, as a function of the magnetic flux density, *B*, for different values of the force of uniform compression as a parameter.

**Figure 10 micromachines-14-01113-f010:**
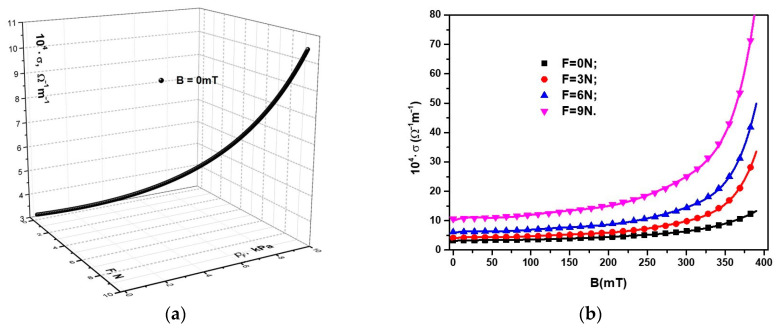
(**a**) The electrical conductivity of the MC, σ, as a function of the compression pressure, $p_{F}$, and the compression force, *F*; (**b**) the electrical conductivity of the MC, σ as a function of the magnetic flux density, *B*, for different values of the force of uniform compression as a parameter.

**Figure 11 micromachines-14-01113-f011:**
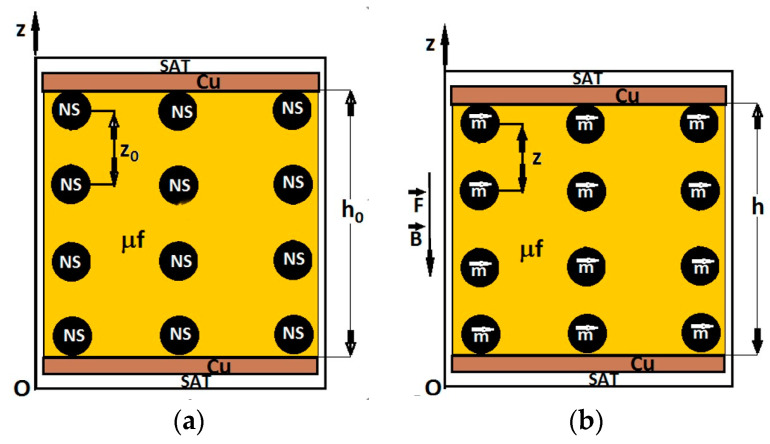
Cross-section through the ED device: (**a**) In the absence of a compression force and a magnetic field; (**b**) in the presence of a compression force and a magnetic field (NS—magnetite micro-ball; $z_{0}$—initial distance between the mass centers of the NS; $\vec{m}$ —magnetic dipole; z —distance between the mass centers of the magnetic dipoles; μf—microfibers of cotton; $h_{0}$ and *h*—thicknesses of MC before and after applying the magnetic field and compression forces, respectively; Oz—coordinate axis; Cu—copper electrodes; SAT—self-adhesive tape; $\vec{B}$ —magnetic flux density vector; $\vec{F}$ —compression force.

**Figure 12 micromachines-14-01113-f012:**
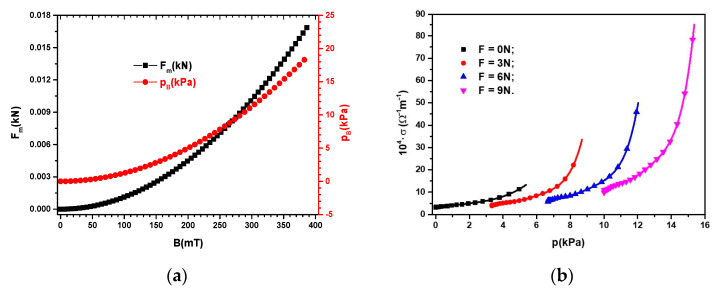
(**a**) The magnetic force, $F_{m}$, and the magnetic pressure, $p_{B}$, depending on the magnetic flux density, *B*; (**b**) the electrical conductivity, *σ*, depending on the total pressure, $p=p_{F}+p_{B}$, with *F* as a parameter.

**Table 1 micromachines-14-01113-t001:** Volumes (*V*) and volume fractions (*Φ*) of the components of MC.

Sample	$V_{f}\left({\mathrm{cm}}^{3}\right)$	$V_{\mathrm{oil}}\left({\mathrm{cm}}^{3}\right)$	$V_{Fe_{3}O_{4}}\left({\mathrm{cm}}^{3}\right)$	$\Phi_{f}\;\left(\mathrm{vol}.\%\right)$	$\Phi_{\mathrm{oil}}\;\left(\mathrm{vol}.\%\right)$	$\Phi_{Fe_{3}O_{4}}\left(\mathrm{vol}.\%\right)$
MC	0.039	0.300	0.021	11	83	7

**Table 2 micromachines-14-01113-t002:** Quantitative analysis of the component elements in the cotton microfibers.

Element	$\Phi_{m}\;\left(wt.\%\right)$	$\Phi_{At}\;\left(At.\%\right)$
C	58.57	69.64
O	30.80	27.53
Al	0.25	0.13
Ca	0.26	0.09
Fe	10.12	2.59

where $\Phi_{m}\;\left(\mathrm{wt}.\%\right)$ is the mass fraction and $\Phi_{\mathrm{At}}\;\left(\mathrm{At}.\%\right)$ is the atomic fraction of the elements in cotton microfibers, respectively.

**Table 3 micromachines-14-01113-t003:** Examples of electric conductivity variations for different values of compression force and density of the magnetic field, *B*.

*F* (N)	$B\left(mT\right)$	$10^{4}\times \sigma \;\left(\Omega^{-1}m^{-1}\right)$
0	0	3.229
	100	3.602
	300	3.458
3	0	4.167
	100	4.705
	300	9.779
9	0	10.600
	100	12.000
	300	24.000

## Data Availability

The data are available from the authors upon reasonable request.

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
