# Peer review of "A Cotton Fabric Composite with Light Mineral Oil and Magnetite Nanoparticles: Effects of a Magnetic Field and Uniform Compressions on Electrical Conductivity"

_micromachines, 2023, doi:10.3390/mi14061113_

Round 1

Reviewer 1 Report

The submitted manuscript is entitled A cotton fabric composite with light mineral oil and magnetite nanoparticles: Effects of a magnetic field and uniform compressions on electrical conductivity.

The article presents an approach for creating low-cost magneto-tactile sensors using a composite material made of cotton fabric impregnated with magnetic liquid. The article describes the experimental setup used to measure the electrical resistance of an electrical device placed in a magnetic field in the absence or presence of uniform compression. The effect of uniform compression and the magnetic field induces mechanical-magneto-elastic deformations and variations in electrical conductivity.

The research topic potentially deals with the field of biomedical engineering, where tactile sensors are increasingly being used. Overall, the article presents an approach for developing magneto-tactile sensors using low-cost and widely available materials, with potential for further research and development. The presented results find well within the scope of the issue of Recent Advances in Magnetoelectric Materials and Devices.

The manuscript requires a revision considering the following issues:

The readability of the manuscript can be improved by moving some parts of the content between sections. At the beginning of the Results and Discussion section, information on an instrument should be moved to the Materials and Methods section. The same is for an optical microscope in the caption of Figure 3.

Conclusions require reorganization. References to Figures and their detailed descriptions should be rather included in the Results and Discussion section.

It seems unnecessary to repeat some information, e.g., on the type of ohmmeter (lines 188, 196). If this information is relevant to the results, it should be placed in a separate section.

Generally, the descriptions of the instruments and methods should be reorganized to improve readability.

There is no description of the light mineral oil in the manuscript, which was used in the present research.

Could the distribution of nanoparticles affect the studied effects? Were the particles evenly distributed in the material? This should be discussed.

Please go through the manuscript and check for typos, e.g.:

Line 108: there should be “in Figure 2a”.

Line 138: there should be “a scanning electron microscope”.

Some minor editing is required.

Reviewer 2 Report

In this work, the authors present the study of electrical properties of cotton fiber-based magnetic composites, which showed very sensitive responses to mechanical stress and magnetic field and thus could potentially provide a low-cost route for magneto-tactile sensors. However, a few critical questions are remaining unclear in the current manuscript that require major revisions before publication:

·        In another paper published by the same group (doi: 10.3390/ma16083222), the authors looked into quite similar experimental setup but the response of electrical resistance to mechanical forces and magnetic field is much weaker than the data reported here, could the authors explain what caused this significant difference?

·        According to equation (5), the magnetic pressures increases with B in a square relationship, but the actual data dependence of electrical conductivity on B looks much steeper than B square (Figure 10(b) and 12(b)), have the authors tried to fit the data to see whether it’s really B square relationship?

·        The authors proposed that the electrical conductivity increases with increasing magnetic field and external force because “the height of the electrical potential barrier decreased with an increase in pressure generated by the compressive force field”, and the impact of magnetic field and compression force field are superimposed on each other. Have the authors tried to find a way (other than simply additive) to unify the impact of magnetic and mechanical pressure? In other words, is there a way to combine pressure from these two origins that can unify data in Figure 12(b) for different forces and magnetic fields into a single, universal curve?

·        The calculation around line 162 is very confusing - it seems that it is implicitly assumed that V_GB = V_MC = 0.360 cm^3 and the calculation is just running in circles around this assumption. Please help clarify the calculation.

·        In line 162 S_MC should have the unit of cm^2.

Round 2

Reviewer 1 Report

The authors appropriately revised the manuscript.

Reviewer 2 Report

The authors provided great feedback on my questions and addressed all my concerns. I don't have any further comments.